# Exosomes in Immune Regulation

**DOI:** 10.3390/ncrna7010004

**Published:** 2021-01-08

**Authors:** Heidi Schwarzenbach, Peter B. Gahan

**Affiliations:** 1Department of Gynecology, University Medical Center Hamburg-Eppendorf, 20246 Hamburg, Germany; 2Department of Tumor Biology, University Medical Center Hamburg-Eppendorf, 20246 Hamburg, Germany; 3Fondazione “Enrico Puccinelli” Onlus, 06126 Perugia, Italy; peterbgahan@gmail.com

**Keywords:** immune therapy, immune cells, extracellular vesicles, non-coding RNAs, microRNAs

## Abstract

Exosomes, small extracellular vesicles mediate intercellular communication by transferring their cargo including DNA, RNA, proteins and lipids from cell to cell. Notably, in the immune system, they have protective functions. However in cancer, exosomes acquire new, immunosuppressive properties that cause the dysregulation of immune cells and immune escape of tumor cells supporting cancer progression and metastasis. Therefore, current investigations focus on the regulation of exosome levels for immunotherapeutic interventions. In this review, we discuss the role of exosomes in immunomodulation of lymphoid and myeloid cells, and their use as immune stimulatory agents to elicit specific cytotoxic responses against the tumor.

## 1. Introduction

A variety of extracellular vesicles (EVs) are released from both healthy and cancer cells having diameters ranging from 20–30 nm to 10 µm; most of them are smaller than 200 nm. Exosomal vesicles (ExVs) are the better studied subgroup of EVs that can be identified by their size of 40–120 nm. In addition, there are some ExV markers used by various scientists as confirmatory indications including Alix, Tsg101, tetraspanins (CD81, CD63, CD9) and flotilla [1]. They have been shown to be present among others in plasma [2], bronchoalveolar lavage [3], breast milk [4], malignant effusions [5], urine [6] and nasal lavage [7].

Mathieu et al. [8] have questioned the ease with which ExVs can be distinguished from other EVs in terms of their biogenesis and functions. The main characteristics of EVs are the transmission of information to acceptor cells either by delivering their content to the recipient cell e.g., fibroblast-derived CB81^+^ by breast cancer cells [9] or by acting at the cell surface with no content delivery e.g., during an immune response with T lymphocyte activation [10]. Within the EVs there is a subgroup of particles, the ExVs that also appear to have a prime function of intercellular communication. As well as general intercellular communication between healthy cells, there is also transmission of molecules involved with tumor induction that are released from existing tumor cells and directed to healthy recipient cells. 

A variety of methods has been employed for the isolation and purification of ExVs by different workers. Each methodological approach is geared to a specific ExV isolation and includes differential centrifugation, density gradient centrifugation, size exclusion chromatography, filtration, polymer based precipitation and isolation by sieving [11], isolation chips [12] as well as immunological separation using a wide range of antibodies with both magnetic beads [11] and magnetic nanowires [13]. Lim et al. [13] point out that ultracentrifugation is a popular (gold standard) method of isolation, but that it is both labor-intensive and time-consuming so leading to a low ExV yield and purity. 

To emphasize the difficulty in choosing a method, a number of examples is given concerning the variety of situations and methods employed. Thus, Shu et al. [14] used cell lines 2183-HER4—a BRAF wild-type melanoma cell line and 888-mel—a BRAF V600E mutant melanoma cell line. They compared ultrafiltration and size exclusion chromatography with ultracentrifugation and found that the former gave 58 times as many ExVs as the latter. A further bonus was the extreme reduction in the amount of contaminating soluble factors. 

In another study by Le Gall et al. [15], adult human primary myoblasts from a muscle biopsy were found to senesce in long-term tissue cultures and needed to be analysed during a highly controlled pre-senescent division count of <21 divisions. ExV isolation from them is handicapped by the low number of cells available from a single muscle biopsy. These workers examined the ExV yield obtained by ultracentrifugation and compared it with a modified polymer-based precipitation strategy employing extra washing steps. The latter was shown to extract ExVs more efficiently than the ultracentrifugation method. Adequate amounts of ExVs were obtained for experimental purposes using lower numbers of differentiated myoblasts than are needed by the ultracentrifugation method [15].

Magnetic micro-beads coated with an antibody to an ExV surface marker, have been exploited in the purification of ExVs. Thus, Clayton et al. [16] isolated ExVs from cell-free culture supernatants using magnetic beads, coated with monoclonal antibodies specific for HLA DP, DQ, DR. Hence, such B-lymphocytes ExVs so derived allowed the demonstration of the expression of MHC Class I and II molecules. Whilst this approach gives relatively pure ExV preparations, a more recent development using magnetic nanowires can significantly improve the isolated sample size and purity. Thus, Lim et al. [13] have extended the magnetic approach by use of a nanowire covered with magnetic particles to each of which is attached streptavidin via ss-biotin. The relevant biotinylated antibody i.e., either CD9 or CD63 or CD81 is then attached to the streptavidin and a magnetic field from 70 to −70 kOe is applied. The processing time is 1h and the minimum volume of human plasma involves only 250 µL. This approach permitted a good harvest of ExVs from human MDA-MB-231 and MCF7 breast cancer cells, HCT116 colon cancer cells, and HeLa cervical cancer cells. Given that this approach can give an approximately threefold greater yield as compared to the above conventional techniques, the nanowires could provide a useful tool in the isolation of ExVs from a liquid biopsy.

Thus, it is important to select a particular method which is suitable for the ExVs in a particular situation e.g., extraction from either a culture medium or a tissue. A helpful account of the advantages and disadvantages of the various methods is given by [11].

## 2. Exosome Biogenesis and Function

ExV formation occurs within the lysosomal system and specifically within multivesicular bodies (MVBs). MVBs develop from late endosomes through membrane in-folding to form small vesicles within, known as intra-vesicular vesicles (IVLs), so leading to the whole structure becoming an MVB. The ExVs are formed via a variety of mechanisms with either one or all mechanisms occurring in the same cell. The best-studied method is one in which MVB formation is coordinated by ESCRT (endosomal sorting complex required for transport). This is comprised of four soluble multi-protein complexes, namely ESCRT-0, ESCRT-I, ESCRT-II and ESCRT-III. Usually, they are associated with the cytosolic side of the endosomal membrane so that particular proteins are sorted into the ILVs [17]. A second approach involves ExV formation in the absence of ESCRT [18]. The precise mechanism for ExV release from the MVBs at the plasma membrane has yet to be fully established [8]. Similarly, three possible mechanisms for miRNA uptake from ExVs have been suggested [19,20]. These involve either membrane fusion followed by release of RNA/DNA or macro-pinocytosis or receptor/raft-mediated endocytosis. 

Thus, ExVs are viewed as an important subgroup of EVs, moving material and informative molecules between cells and over varying distances. Given the wide range of molecules present, it is clear that, the recipient cells can have their biology modified [21].

ExVs contain a variety of molecules including both genomic and mitochondrial DNA (100 bp–17 kb). Emphasis has been placed on ExV DNA concerning the identification of the presence of a particular cancer and the timing of the appearance of metastases via the liquid biopsy procedure [22,23]. However, it can also be observed that the movement of DNA from cancer to healthy recipient cells can influence the biology of the recipient cells leading to tumor formation [21]. Nevertheless, the literature on the estimation of the DNA content of ExVs is somewhat confusing in that the presence of 93% of the DNA is reorted to be present in ExVs, the remaining DNA being present in the plasma and often being found attached to the outer surface of the ExVs. In contrast, it was shown that there no DNA is present in ExVs, but is present in the larger EVs emanating from tumor cells [24,25].

Additional and important components of ExVs are RNAs, including mRNAs, rRNAs (ribosomal RNAs, tRNAs (transfer RNAs), miRNAs (microRNAs), shRNAs (short hairpin RNAs), lncRNA (long noncoding RNAs), piRNA (piwi-interacting RNAs), snRNA (small nuclear RNAs), and proteins, including ceramides and cholesterol [26]. The mitochondrial DNA and RNAs are present together with mitochondrial proteins that can count for as much as 10% of total ExV proteins [27]. The latter can include the electron transport system as well as complete mitochondria [28]. The ExV RNA and protein contents are of particular interest as they include miRNAs, shRNA, lncRNAs and components of the immune system.

ExVs become especially important when considering the movement of cell signals from tumor cells to healthy cells when they can help cancer cells to propagate genetic information that leads to the development and maintenance of metastases. Equally, such ExVs may be exploited in the treatment of tumors by acting as carriers of anti-tumor compounds.

## 3. Exosomal RNAs

ExVs contain varying amounts of RNAs concerning over a dozen different RNA forms [21], the majority of which are classified as ncRNAs (non-coding RNAs). They do not normally code for protein synthesis, but they are able to influence mRNAs in either stabilizing mRNAs, so serving to protect mRNA from degradation, or degrade mRNA. Nevertheless, it is important to know whether or not any mRNA molecules present are intact since they will encode for functional proteins and hence transfer functions of the original cell into the recipient cell. Moreover, partially degraded molecules could deliver new traits into the target cell [29]. Of the ncRNAs, perhaps the two most important in current ExV cancer studies are miRNAs and lncRNAs. Given the possibility of using ExVs as carriers of molecules to attack tumors, e.g., shRNAs, a brief resume of the RNAs present in ExVs is given.

### 3.1. Natural Occurring RNAs

#### 3.1.1. miRNAs

miRNAs are produced from an initial RNA strand derived from a host gene involving typical splicing, capping and polyadenylating so leading to the development of mature and active 21–23 nucleotide miRNAs [30,31]. The latter integrate with an RNA-induced silencing complex (RISC) that targets the mRNAs to be degraded or inhibited [21]. This makes them useful markers that are present in liquid biopsies and so for the monitoring of the presence, treatment and reappearance of a particular cancer type.

#### 3.1.2. Sponge circRNAs

Sponges may be formed from both linear and circular RNAs (circRNA). CircRNAs are a class of ncRNAs that are formed by 3′-5′ ligation of an RNA molecule. There are three forms of circRNA depending upon the nuclear origin, i.e., whether formed from introns (ciRNA) or exons (ElciRNA) or exon-introns (ecRNA). CircRNAs are important in the regulation of miRNAs by acting as “miRNA sponges” in e.g., the presence of cancer [32,33]. Primarily found in the cytosol, circRNAs—mainly CIRS-7 and SrycircRNA are involved with the latter having 16 binding sites for miR-138 [34]. CDR1as/CiRS-7 is encoded in the genome antisense to the human locus resulting in the name CDR1. This acts as a sponge having over 60 binding sites of miR-7 (CiRS-7—circular RNA sponge for miR-7) [35]. MiR-7 is also pulled down by circHIPK3 in colon cancer cases [36]. Thus, a mechanism exists to reduce the number of e.g., miR-7 in the cell. However, if such RNA molecules are rapidly required in numbers by the cell, they could be released from the sponge. This possibility is based upon CiRS-7 being sliced by miR-671 so leading to a possibility for the existence of a system to release miRNAs at an appropriate time [37]. Zhang et al. [38] have demonstrated that circTRIM33-12 can act as a sponge for miR-19 in hepatocellular carcinoma (HCC). Downregulated circTRIM33-12 was found to upregulate TET1 (ten-eleven translocation methylcytosine dioxygenase 1) expression on sponging miR-191. However, downregulation of circTRIM33-12 in HCC was significantly correlated with malignant characteristics affecting both overall survival and recurrence-free survival after surgery. This basic prognosis appears to be due to the circTRIM33-12 sponging miR-191 and upregulating TET1 expression and hence, leading to significantly reduced 5-hydroxymethylcytosine levels in HCC cells.

Until now, very few circRNAs have been shown to have sponge characteristics [39,40]. 

#### 3.1.3. lncRNAs

lncRNAs are a class of RNA having more than 200 nucleotides though lacking the potential to code for proteins [41,42]. lncRNAs play significant roles in gene regulation and expression and participate in a variety of biological processes that include imprinting, cell growth, apoptosis and differentiation [41,42]. Their expression levels become dysregulated in patients having different cancer types in association with tumorigenesis, cancer progression and metastases [43]. 

More than 210,000 lncRNAs are present [44], 106,063 being human related (LncRNAWiki, 2015) though only 14,470 were recorded in Gencode (2014). Of these, only 1867 human lncRNAs appeared to be biologically active in humans [45]. Although there is a very large number of lncRNAs that have been identified, very few have been considered for ExV lncRNA involvement in cancers and cancer resistance to treatment. For example, only HOTAIR, AGAP2-AS1, SNHG14 and UCA1 have been considered for breast cancer resistance to tamoxifen and trastuzumab [43,46,47,48,49].

### 3.2. Artificial miRNAs

#### shRNAs

shRNAs are artificially created molecules with a tight hairpin turn that gives them the capacity to silence target gene expression via RNA interference (RNAi). They are usually delivered by through plasmid delivery though they are suitable as an exosome component to attack tumor cells. Initially, the cytosolic delivery of RNAi oligonucleotides was only used with cells capable of transfection and, in particular, transient in vitro studies. With the advent of its use in gene function studies, the need for a more reliable mechanism of delivery was needed. An improved method involved the development of shRNA which permitted infection with viral vectors leading to stable integration of shRNA and long-term knockdown of the targeted gene [50].

shRNAs integrate into the DNA by virtue of two complementary 19–22 bp RNA sequences linked by a short loop of 4–11 nucleotides that resembles the hairpin occurring naturally in miRNAs. After transcription, the shRNA sequence moves to the cytosol where it is processed by the Dicer system as is the case with miRNAs. Subsequently, the siRNA so produced binds to the relevant mRNA and the complex moves to the RISC where the mRNA is broken down [51].

## 4. Exosomal Proteins

A major component of ExVs concerns proteins linked to the immune system. These may be secreted from healthy cells yielding a defense mechanism against tumors. They may also present a first line of nasal infection defense as demonstrated by Lasser et al. [7] showing the induction of nasal immune cell migration by nasal ExVs. Furthermore, their presence in breast-milk may also implicate them in the development of an infant’s immune system [4]. 

They are also released by tumor cells as shown by early studies implicating the involvement of the immune system proteins and the implications for tumor suppression [52]. Moreover, they can have an impact on tumor cell survival e.g., tumor-derived ExV PD-L1 suppression of T cell activation in the draining lymph node [53]. In addition, these authors indicate that the ExV PD-L1 would seem to be resistant to anti-PD-L1 antibody blockade.

This review will concern especially the involvement of the ExVs in the immune system—those cells are shown in Figure 1 to suppress tumor cell survival. In addition, the manner in which ExVs can be used as vehicles to deliver immune factors to block tumor cells will be considered.

Table 1 summarizes the ExVs carrying miRNAs, lncRNAs or PD-L1. Their function in immune regulation mentioned in this table is described in detail in the following paragraphs.

AML, acute myelogenous leukemia; BE, Barrett’s esophagus; CAFs, cancer-associated fibroblasts, CCL2, C-C motif chemokine ligand 2; DCs, dendritic cells; CLL, chronic lymphocytic leukemia; DNMT3A, DNA methyltransferase 3A; EAC, esophageal adenocarcinoma; EZH2, histone methyltransferase enhancer of zeste homolog 2; EED, EZH2 cofactor embryonic ectoderm development; EGCG, epigallocatechin gallate; FGF11, fibroblast growth factor 11; GBM, glioblastoma multiforme; GC, gastric cancer; Hbp1, high-mobility group box transcription factor 1; HCC, hepatocellular carcinoma; HNC, head and neck cancer; HNSCC, head and neck squamous cell carcinoma; LLC, LRRC4, leucine rich repeat containing 4; MARK1, microtubule affinity regulating kinase 1; MDSCs, myeloid-derived suppressor cells; MMP 9, matrix metallopeptidase 9; NK, natural killer; NF-κB, nuclear factor kappa B; NKG2D, natural killer group 2 member D; NPC, nasopharyngeal carcinoma; NSCL, non-small cell lung; OSCC, oral squamous cell carcinoma; Prkar1a, protein kinase cAMP-dependent type I regulatory subunit alpha; TAMs, tumor-associated macrophages; RORA, RAR-related orphan receptor alpha.

## 5. Exosomes and Lymphoid Cells

### 5.1. T Cells

The initiation of antitumor immune responses mediated by lymphoid T cells involves the uptake and processing of tumor antigens by monocyte-derived dendritic cell (DCs) and their presentation on major histocompatibility complex class I (MHC-I) molecules [54]. In 2001, Wolfers et al. [55] showed that ExVs constitute a pool of tumor-rejection antigens for T-cell cross priming, relevant for immune interventions. In their in vitro model system, ExVs secreted by tumor cells presented and transferred tumor antigens to DCs. After uptake, DCs induced potent cytotoxic CD8^+^T-cell-dependent antitumor effects on syngeneic and allogeneic established mouse tumor models. In 2011, Mittelbrunn et al. [56] showed that ExVs of T cells, B cells and DCs contain miRNA patterns that differ from those of their parent cells. They demonstrated the existence of antigen-driven unidirectional transfer of miRNAs from the T cell to antigen presenting cells (APCs) mediated by CD63^+^ExVs on immune synapse formation. Recently, Hong et al. [57] developed an acute myelogenous leukemia (AML) patient-derived tumor xenograft model in mice. Engrafted mice produced ExVs that carried human proteins and leukemia-associated antigens, similar to those of ExVs isolated from the plasma of AML patients who had donated the cells for engraftment. This raises the possibility that AML blasts are the source of immunosuppressive ExVs with a molecular profile that mimics the content and functions of the parental cells. Incidentally, treatment of tumor cells with the chemotherapeutic agent cyclophosphamide was reported to be a useful method to (a) enhance the secretion of these ExVs in sensitive cell lines without altering their β-activator of DCs and (b) change the morphological characteristics of DCs upon contact with non-cognate activated bystander T-cells. Non-activated bystander T-cells had no such effects. Stimulation of DCs with activated bystander T-cells increased the release of ExVs containing miR-155 that is known to be a central modulator of T-cell responses. These interactions between DCs and bystander T-cells modulated antigen-specific T-cell responses via ExVs [58]. Furthermore, in the tumor microenvironment, mesenchymal stem cells (MSCs) are important components in the activation of immune cells and the maintenance of an inflammatory environment [59]. In their cell culture experiments, Shen et al. [60] demonstrated that MSCs treated with ExVs, derived from a gastric cancer (GC) cell line, upregulated the secretion of pro-inflammatory factors, activated CD69 and CD25 on the T cell surface and provoked macrophage phagocytosis. These immunomodulatory effects were mediated by the abnormal activation of the Nuclear Factor kappa B (NF-κB) signaling pathway [61] in MSCs after GC ExV administration,. This was due to the inhibition of the NF-κB signaling pathway that markedly weakened the effect of MSCs stimulated by GC ExVs on T cells and macrophages.

Immune escape may be caused by the programmed death ligand 1 (PD-L1). This type I transmembrane protein on the surface of tumor cells binds to its T cell receptor PD-1 to suppress their activation. Conversely, inhibition of PD-L1 by specific antibodies can induce an anti-tumor immune response [62]. Unfortunately, resistance to current anti-PD-L1 antibody approaches has been identified. Recently, Poggio et al. [53] detected that removal of PD-L1-bearing ExVs inhibited tumor growth, even in mouse models resistant to anti-PD-L1 antibodies. In the draining lymph node, PD-L1-bearing ExVs released from the tumor suppressed T cell activation. Exposure to ExV PD-L1-deficient tumor cells suppressed growth of "wild-type” tumor cells injected at a distant site. Notably, the tumor-suppressive effect of anti-PD-L1 antibodies was additive and not redundant. In addition, Kim et al. [63] found that PD-L1 was also present on the surface of ExVs isolated from the plasma of non-small cell lung (NSCL) cancer patients and that its abundance on ExVs was associated with the occurrence of PD-L1 in tumor tissues. PD-L1-expressing ExVs secreted from lung cancer cells reduced the secretion of IFN-γ by Jurkat T cells, enhanced tumor growth in vivo and induced apoptosis in CD8^+^T cells. IFN-γ secretion and immune functions could be restored by PD-L1 knockout or masking of the ExVs. Finally, Chen et al. [64] described the presence of PD-L1 on metastatic, melanoma-released ExVs. Conversely, stimulation with IFN-γ elevated the yield of PD-L1 on these ExVs so suppressing the function of CD8^+^T cells and facilitating tumor growth. The levels of circulating ExV PD-L1 positively correlated with that of IFN-γ in metastatic melanoma patients, but varied during the course of anti-PD-1 therapy and could stratify clinical responders from non-responders. An inhibitory effect on GC progression through activation of T cell immune response via PD-L1 was shown by Li et al. [65]. HCC cells released ExVs containing PCED1B-AS1 that enhanced the expression of PD-L1 and PD-L2 in recipient HCC cells whereas they inhibited receipt T cells and macrophages. The levels of ExV PCED1B-AS1 in blood correlated with the expression of PD-L1 and PD-L2. PCED1B-AS1 promoted cell proliferation, colony formation and inhibited apoptosis. Finally, tumor formation was promoted in xenografted nude mice. A significantly high level of miR-92 in ExVs of cancer-associated fibroblasts [CAFs] was detected in breast cancer by Dou et al. [66]. Following treatment by CAF-derived ExVs, breast cancer cells expressed higher levels of PD-L1, accompanied with increased miR-92 expression. Increased PD-L1 expression, that was induced by CAF-derived ExVs, significantly promoted apoptosis and impaired proliferation of T cells. LATS2 was identified as a target gene of miR-92. LATS2 can interact with YAP1 that, after nuclear translocation, could bind to the enhancer region of PD-L1 to promote transcription activity. NSCLC patients harbor unique plasma ExV miRNA profiles. MiR-320d, miR-320c and miR-320b were identified as potential biomarkers for predicting the efficacy of immunotherapy in advanced NSCLCs. During the treatment, a T-cell suppressor, miR-125b-5p, was downregulated so leading to increased T-cell function and a good response to immunotherapy by these patients [67].

Wu et al. [68] described dysregulated ExVs containing RNA and proteins that may involve in the development of NSCLC. Higher levels of ExVs are found in the lungs of smokers and NSCLC patients. Numerous dysregulated miRNAs were detected in NSCLC patients when compared to smokers. Bioinformatic analysis demonstrated that proteoglycans, fatty acid biosynthesis, ErbB, Hippo, TGF-beta, Wnt, Rap1, AMPK and Ras pathways were the most prominent pathways enriched in ExV miRNA signatures. ExVs from both smokers and NSCLC patients contained high levels of lncRNAs, including MALAT1, HOTAIR, HOTTIP, AGAP2-AS1, ATB, TCF7, FOXD2-AS1, HOXA11-AS, PCAF1, and BCAR4. Xu et al. [69] found longitudinally decreased expression levels of miR-15a, miR-15b, miR-16, and miR-193a-3p in ExVs during the development from Barrett’s esophagus (BE) status to esophageal adenocarcinoma (EAC) progression. The ExV miRNAs that potentially targeted PD-L1 mRNA were down-regulated in PD-L1-positive BE and EAC patients.

Toll-like receptor 4 (TLR4) is also involved in the initiation of innate and adaptive immune responses, its increased activity in chronic infectious and inflammatory diseases contributing to the pathogenesis of cancer [70]. Thus, Domenis et al. [71] investigated the immune modulatory properties of immunosuppressive ExVs secreted by tumor cells upon TLR4 activation when tumor cells escaped immune surveillance, so supporting tumor progression. They detected that glioma-derived ExVs suppressed the T-cell immune response by acting on monocyte maturation rather than by direct interaction with T cells. Peripheral blood mononuclear cells (PBMCs) of healthy donors stimulated with anti-CD3, anti-CD28 and IL-2 were treated with glioma-stem cell (GSC)-derived ExVs. In unfractionated PBMCs, GSC-derived ExVs inhibited T cell activation, proliferation and T helper 1 (Th1) cell cytokine production. In PBMC culture, glioma-derived ExVs directly promoted IL-10 and arginase-1 production [72]. Hellwinkel et al. [73] incubated PBMCs with mitogenic stimuli and various concentrations of glioma-derived ExVs. Functional analyses revealed a decreased migratory capacity of PBMCs after incubation with these ExVs as well as differential effects of high and low ExV concentrations. ExVs at high concentrations induced selective tolerance to an immune response. As previously assumed, lymphocytes in patients’ circulation were not irreparably impaired, but could be rescued to elevate antitumor responses. However, Iorgulescu et al. [74] suggested that glioma-derived ExVs are restricted in their capacity to directly prime peripheral immunosuppression. Although treatment with glioma-derived ExVs promoted immunosuppressive human leukocyte antigen-DR (HLA-DR)^low^ monocytic phenotypes, it failed to either induce monocytic PD-L1 expression or alter the activation of cytotoxic T-cells. In another study on glioma, Mirzaei et al. [75] established a novel immunosuppressive role for the extracellular matrix protein tenascin-C (TNC), which is associated with ExVs secreted by stem-like brain tumor-initiating cells (BTICs), to affect local and distal T lymphocyte immunity. They found that circulating ExVs from glioblastoma patients contained more TNC and T cell-suppressive activity than those from control individuals. Using co-cultures, they presented evidence that BTICs released ExVs containing TNC and that, in turn, TNC inhibited T cell proliferation through interaction with the integrins α5β1 and αvβ6 on T cells so reducing mTOR signaling. In addition, TNC-depleted ExVs suppressed T cell responses to a significantly lesser extent than their controls. Finally, Li et al. [76] described a new strategy to re-express the tumor suppressor gene, leucine rich repeat containing 4 (LRRC4), in glioblastoma multiform cells. LRRC4, the expression of which is important for the development of the nervous system, is hypermethylated and consequently downregulated in glioma [77]. Li et al. [76] showed that GBM (glioblastoma multiforme) cell-derived ExVs containing programmed cell death 1 (PD1) contributed to the modulation of LRRC4 on T cells, whereby miR-101 reversed the promoter hypermethylation and induced the re-expression of LRRC4 in GBM cells by targeting the histone methyltransferase enhancer of zeste homolog 2 (EZH2), the EZH2 cofactor embryonic ectoderm development (EED) and DNA methyltransferase 3A (DNMT3A).

For the first time, Zhou et al. [78] revealed that melanoma-released ExVs activate the mitochondrial apoptotic pathway [79] in CD4^+^T cells. Tumor-derived ExVs were able to increase the activation of caspase-3, caspase-7 and caspase-9 and down-regulate the anti-apoptotic proteins, including BCL-2, MCL-1 and BCL-xL, in CD4^+^T cells. The subsequent reduction in T cells may be caused by, amongst other things, the release of ExV miR-690 in the T cells so resulting in the accelerated growth of melanoma cells in mice [78]. In this regard, Bland et al. [80] showed that ExVs derived from melanoma cells were enriched with coding and noncoding RNAs that did not reflect the abundance in their parental cells. Upon delivery, melanoma-derived ExVs induced dynamic changes in the transcriptome of cytotoxic T cells leading to altered mitochondrial respiration and upregulated genes associated with the Notch signaling pathway. The same laboratory [81] also examined ExVs derived from three melanoma-related cell lines. Interestingly, only those derived from the melanoma B16F0 cell line contained both protein and mRNA for protein tyrosine phosphatase non-receptor type 11 (PTPN11) which inhibited T-cell proliferation.

Tregs are a subpopulation of T cells that maintain tolerance to self and limit other immune responses [82]. For the first time, Tung et al. [83] described the transfer of miRNAs from Tregs to DCs by ExVs. The elevated levels of miR-150-5p and miR-142-3p in DCs that were initiated by their interaction with Treg-derived ExVs led to the acquisition of a tolerant phenotype in these cells with increased IL-10 and decreased IL-6 production following LPS stimulation. The ratio of Tregs to Th17 cells was measured by Zhou et al. [84]. The ratio of Tregs/Th17 cells was found to be significantly higher in both epithelial ovarian cancer [EOC] in situ and metastatic peritoneal tissues than in benign ovarian tumors and benign peritoneum. They further identified miR-29a-3p and miR-21-5p to be enriched in ExVs derived from tumor-associated macrophages [TAMs]. Transfection of CD4^+^T cells with both miRNAs suppressed signal transducer and activator of transcription (STAT) 3 and regulated Tregs/Th17 cells so indicating that ExVs mediated the interaction between immune cells. This exosomal mediated interaction between T cells and TAMs generated an immuno-suppressive microenvironment and facilitated EOC progression and metastasis. Moreover, in a mouse model, Xie et al. [85] showed that ExVs secreted by CD8^+^ and CD25^+^Tregs were able to suppress cytotoxic T lymphocyte-mediated immunity against melanoma immunized mice. Interestingly, Guo et al. [86] revealed that ExVs derived from heat-stressed tumor cells were able to convert Tregs into Th17 cells with a high efficiency, a process that was dependent on IL-6, and consequently exerted strong antitumor effects.

A small population of T lymphocytes constitutes γδ T cells which exert either anti- or pro-tumoral functions in cancer [87]. Li et al. [88] investigated whether tumor-derived ExVs influence the anti- and pro-tumoral equilibrium of γδ T cells under different oxygen pressures in the tumor microenvironment. They showed that tumor-derived ExVs were able to support the expansion and cytotoxicity of γδ T cells in a DC-independent manner. The stimulatory effect of normoxic tumor-derived ExVs on γδ T-cell activity was not observed with hypoxic tumor-derived ExVs that, in contrast, increased the suppressive effect of myeloid-derived suppressor cells (MDSCs) on γδ T cells through regulating the miR-21/PTEN/PD-L1 axis. In particular, the therapeutic outcome benefited from combined miR-21 and PD-L1 targeting in an oral squamous cell carcinoma (OSCC)-bearing immunocompetent mouse model. These findings indicate that different oxygen pressures in the tumor microenvironment may change the ExV content.

Using anti-CD3 antibodies, Theodoraki et al. [89] separated CD3^+^ from CD3-ExVs. Surprisingly, the mean levels of PD-L1, cytotoxic T lymphocyte antigen 4 (CTLA-4) and cyclooxygenase-2 (COX-2) were similar in both ExV subtypes. ExVs from patients with Union for International Cancer Control (UICC) stages III/IV disease harbored higher levels of ExV PD-L1 and COX-2 than stages I/II patients. Lymph node-positive patients had ExVs with a higher immunosuppressive protein content than lymph node-negative patients. Furthermore, patients with head and neck squamous cell carcinoma (HNSCC), a subtype of head and neck cancer (HNC), displayed ExVs with higher levels of PD-L1, COX-2 and CD15 than did healthy donors. These researchers also investigated the role of both ExV subtypes during immune cell reprogramming by focusing on the adenosine pathway components [90]. The highest level of adenosine production was found in CD3-ExVs in HNSCC patients with stages III/IV. Additionally, the production of 5’-AMP and purines was significantly higher in Tregs co-incubated with CD3-ExVs than those incubated with CD3^+^ ExVs. Consistently, CD26 and adenosine deaminase (ADA) levels were higher in CD3^+^ than in CD3-ExVs. ADA and CD26 levels in CD3^+^ ExVs were also significantly higher in patients with early (stages I/II) than advanced (stages III/IV) disease. This opposite role indicates the specific immunosuppressive character of both adenosine-producing ExV subsets in early versus advanced HNSCC [91]. Notably, Maybruck et al. [92] showed for the first-time that multiple HNC-derived cell lines secreted ExVs that induced a suppressor phenotype in CD8^+^T cells as typified by the loss of CD27/CD28 expression and acquisition of a potent suppressor function. Their mass spectrometric analyses revealed that the soluble glycan-binding protein, galectin-1, (Gal-1) [93] was enriched in those ExVs. In addition, Ludwig et al. [94] showed that ExV-induced immune suppression correlated with disease activity in HNC. ExVs of HNC patients with active disease were significantly more effective in inducing apoptosis of CD8^+^T cells, suppression of CD4^+^T-cell proliferation and upregulation of Treg suppressor functions than those from individuals with no evident disease. ExVs from these patients also downregulated the activating receptor natural killer group 2 member D (NKG2D) [95] in NK cells [94]. Also of note is that human papillomavirus (HPV)-infected HNSCC respond more favorably to therapy than HPV- HNSCC. To check whether or not ExVs produced by HPV^+^ or HPV^−^HNSCC differentially modulate anti-tumor immune responses, Ludwig et al. [96] performed a comparison of proteome profiles of these ExVs by high-resolution mass spectrometry. Only HPV^+^ ExVs were enriched with immune effector cell-related CD47 and CD276 antigens, whereas only HPV-ExVs contained the tumor-protective/growth-promoting antigens MUC-1 and HLA-DA. Hence, the protein content in HPV^+^ and HPV^−^ExVs might contribute to the disparity in immune responses that characterize HPV^+^ and HPV^−^HNSCC.

In several cancer types, overexpression of gangliosides at the outer leaflet of the plasma membrane, such as GD3, leads to activation of cell signaling and increasing cell proliferation and migration [97]. Shenoy et al. [98] detected GD3 on the surface of ExVs isolated from EOC ascites fluids. They found that GD3-mediated deactivation of the T cell receptor was dependent on sialic acid groups since their enzymatic removal from ExVs or even liposomes resulted in a loss of the inhibitory exosomal capacity. Thus, GD3 and sialic acid molecules on the exosomal surface represent potential therapeutic targets for boosting the antitumor activity of quiescent T cells in the tumor microenvironment. The same laboratory [99] also reported that T cells pulsed with tumor-associated ExVs extracted from EOC ascites fluids inhibited the translocation of NFκB and Nuclear factor of activated T cells (NFAT) into the nucleus, upregulation of CD69 and CD107a, production of cytokines and T cell proliferation. However, if the ExVs were removed, T cells could be reactivated.

A frequent site of breast cancer (BC) metastasis is lung. Accordingly, fluorescently labeled ExVs derived from highly metastatic murine BC cells predominantly disseminated into the lung of syngeneic mice. At the sites of accumulation, ExVs were taken up by CD45^+^ bone marrow-derived cells. In this context, Wen et al. [100] performed long-term conditioning of naive mice with ExVs from highly metastatic BC cells leading to the accumulation of MDSCs in the lung and liver, and the immune suppressive microenvironment. In these pre-metastatic organs, BC-derived ExVs suppressed T-cell proliferation and inhibited the cytotoxicity of NK cells. 

The neuropeptide Neuromedin U (NmU) plays a crucial role in HER2-positive BC and correlates with increased aggressiveness, resistance to HER2-targeted therapies and overall poorer patients’ outcome [101]. Martinez et al. [102] showed that HER2-positive BC cells that over-expressed NmU increased resistance to the HER2-specific antibody trastuzumab so indicating a role of NmU in enhancing immune evasion. The BC cells released ExVs with elevated amounts of the tumor growth factor (TGF)-β1 and PD-L1. In their neo-adjuvant clinical trial, the researchers also found that the levels of the immunosuppressive cytokine were significantly higher in ExVs isolated from the serum of HER2-overexpressing BC patients who did not respond to HER2-targeted drug treatment than those who experienced complete or partial response. Apart from the immunosuppressive character of tumor-derived ExVs, Li et al. [103] generated HER2-specific ExVs using polyclonal CD4^+^T cells prepared with ExVs released by HER2-specific DCs, and demonstrated their therapeutic effect against HER2-positive tumors in double-transgenic HER2/HLA-A2 mice with HER2-specific self-immune tolerance. In addition, Xie et al. [104] developed a DNA vaccine composed of heterologous human HER2 and rat neu sequences. These heterologous human/rat HER2-specific ExVs stimulated stronger HER2-specific CD8^+^T-cell responses so eradicating 90% of HER2-specific target cells than did HER2 ExVs that induced CD8^+^T-cell responses, but only eliminated 53% of target cells. Furthermore, vaccination with the heterologous HER2 ExVs was capable of circumventing HER2 tolerance and suppressing lung metastasis. 

ExVs with their carcinogenic cargo are also key mediators in the immune regulation of nasopharyngeal carcinoma (NPC). Ye et al. [105] showed that exosomal miR-24-3p is involved in NPC pathogenesis, mediating T-cell suppression via repression of fibroblast growth factor 11 (FGF11). The markedly elevated levels of miR-24-3p in ExVs isolated from cell lines and patient serum correlated with worse NPC patients’ disease-free survival. Knockdown of ExV miR-24-3p by a sponge RNA restored the ExV-mediated inhibition of T-cell proliferation, Th1 and Th17 differentiation and the induction of Tregs. Administration of ExV miR-24-3p increased the expression of phosphorylated ERK and the phosphorylated transcription factors STAT1 and STAT3, but decreased the expression of phosphorylated STAT5 during T-cell proliferation and differentiation. Of note, the levels of FGF11 which were inhibited by miR-24-3p were positively associated with the number of CD4^+^ and CD8^+^T cells in vivo. Moreover, hypoxia increased the levels of cellular and ExV miR-24-3p and enhanced the inhibitory effect of tumor-derived ExVs on T-cell proliferation and differentiation. The same laboratory [106] also found that overexpression of miR-24-3p, along with miR-891a, miR-106a-5p, miR-20a-5p and miR-1908, down-regulated the microtubule affinity regulating kinase 1 (MARK1) signaling pathway in T cells. In addition, Mrizak et al. [107] demonstrated that NPC-derived ExVs increased the expansion of Tregs, activating the inhibitory receptor T-cell immunoglobulin and mucin domain 3 (TIM3) along with an increased expression of CD25 and forkhead box P3 (FOXP3), a lineage master regulator for Treg development and suppressive activity [82]. This regulation of Tregs led to a stronger suppression of responder cell proliferation and the secretion of IL-10 and TGF-β1 [107]. 

As mentioned above, the impairment of CD8^+^T cell responses also compromises the adenosine pathway [90]. In this regard, Salimu et al. [108] showed that prostate cancer-derived ExVs activated the expression of CD73, an ecto-5-nucleotidase responsible for the hydrolysis of AMP into adenosine, on DCs so resulting in an ATP-dependent inhibition of TNFα- and IL-12-production and suppression of DC function via ExV prostaglandin E2. Conversely, DU145 cells transfected with shRNA to knockdown the GTPase Rab27a, a member of the Rab family that regulates exocytosis of distinct vesicles including ExVs [109], inhibited ExV secretion and triggered significantly stronger tumor-antigen-specific T cell responses when loaded onto DCs than did untransfected prostate cancer cells [108]. In addition, in Tregs, being more sensitive to effects mediated by tumor-derived ExVs than other T cell subsets, ExVs down-regulated genes for the adenosine pathway and increased adenosine production [110].

In contrast, antitumor effects triggered by ExVs have been described for several tumor types. Zhang et al. examined the ability of ExVs derived from glycosyl-phosphatidylinositol-anchored IL-2 (GPI-IL-2) gene-modified bladder cancer cells to increase antitumor responses. They transfected melanoma antigen-1 (MAGE-1)-expressing T24 tumor cells with a plasmid encoding for GPI-IL-2. The ExVs secreted by these cells contained bioactive GPI-IL-2 and tumor-associated antigen MAGE-1. Interestingly, DCs pulsed with these ExVs induced the proliferation of T cells and the antigen-specific cytotoxic T-lymphocyte immune response more efficiently [111]. Furthermore, Xu et al. [112] found that ExVs derived from RCC cells stimulated CD8^+^T cells, and in combination with GM-CSF and IL-12, effectively facilitated cytotoxicity against autologous RCC [renal cell carcinoma] cells in vitro. Immunization with RCC-derived ExVs restrained the growth of RCC tumors in a mouse model as well as facilitating the induction of a stronger specific cytotoxic CD8^+^T cell response via the Fas ligand/Fas mediated apoptotic signaling pathway [113] in vitro. However, these stimulatory effects were specific for the RCC cell line RenCa and less for other tumor cell lines, suggesting that this antitumor immune response may depend on the antigen specificity of RCC-derived ExVs [112]. Also in a mouse model, Yang et al. [114] showed that injection of ExVs released by hepatic cancer cells, either infected with IRF-1-expressing adenovirus or treated with IFN-γ, had improved antitumor effects. This event was the result of increased infiltration of CD4^+^ and CD8α^+^T cells in tumors. Furthermore, splenocytes isolated from these mice had a higher number of IFNγ-positive and granzyme B-positive CD8^+^cells. ExVs isolated from ascites of mice bearing a very aggressive murine T-cell lymphoma were characterized by Menay et al. [115]. The injection of ascites-derived ExVs into naive-syngeneic mice induced both humoral and cellular immune responses that allowed tumor rejection along with Th1 responses. The additional in vitro analysis discovered that T-cells from ascites-derived ExV-immunized mice secreted IFN-γ in response to antitumor stimulation.

### 5.2. B Cells

Antigen-specific interactions between B and T cells are essential for an efficient immune response and require MHC class II complexes on the B cell to interact with the T cell receptor on antigen-specific T cells. In 2007, Muntasell et al. [116] examined the mechanisms that regulates the persistence, loss and secretion of specific MHC-II complexes on activated B cells. They found that activated B cells degraded approximately 50% of MHC-II molecules each day and secreted about 12% of these molecules, previously expressed on the plasma membrane of B cells, by ExVs. This regulated ExV release from activated B cells allows MHC-II to escape intracellular degradation and ExVs to directly stimulate primed, but not naïve, CD4^+^T cells. 

The process through which tumor-associated antigens trigger B cell response in pancreatic ductal adenocarcinoma (PDAC) was investigated by Capello et al. They provided evidence that PDAC-derived ExVs display a large repertoire of tumor antigens that induced a dose-dependent inhibition of PDAC serum-mediated, complement-dependent cytotoxicity towards cancer cells [117].

Regulatory B cells (Bregs) constitute a subset of B cells that infiltrate solid tumors and possess distinct phenotypes in different tumor microenvironments [118]. T cell immunoglobulin and mucin domain 1 (TIM-1), a transmembrane glycoprotein, acts as a marker for Bregs. At first, TIM-1 was assumed only to play a role in the activation of CD4^+^T cells and DCs. Now, it is known that TIM-1 is mainly expressed on B cells [119]. In this regard, Ye et al. [120] described a novel mechanism of TIM-1^+^Breg-mediated immune escape. They detected that hepatocellular carcinoma (HCC) patients displayed a significantly higher TIM-1^+^Breg infiltration in their tumor tissue than in the paired peritumoral tissue. The infiltrating TIM-1^+^Bregs expressed high levels of IL-10 and possessed strong suppressive activity against CD8^+^T cells. B cells activated by HCC-derived ExVs significantly expressed TIM-1 protein and acquired a suppressive character against CD8^+^T cells similar to TIM-1^+^Bregs isolated from HCC tumor tissue. ExV-derived high mobility group box 1 (HMGB1) both activated B cells and promoted TIM-1^+^Breg cell expansion via the TLR 2/4 and MAPK signaling pathways.

The induction of the release of ExVs containing the viral latent membrane protein 1 (LMP1) by Epstein-Barr virus (EBV)-infected B cells was demonstrated by Gutzeit et al. [121]. Constitutive signaling of LMP1, that imitated CD40 signaling and induced proliferation of B and T cell-independent class-switch recombination, was reduced because of the shedding of LMP1 via ExVs. In cell culture experiments, LMP1-carrying ExVs derived from the DG75 Burkitt’s lymphoma cell line were internalized by B cells resulting in proliferation, induction of activation-induced cytidine deaminase (AID), the production of circular and germline transcripts for IgG1 in B cells and drove B cell differentiation toward a plasmablast-like phenotype.

### 5.3. Natural Killer Cells

NK cells and NK-derived ExVs induce tumor cell cytotoxicity by activating the cell killing proteins perforin and granzyme. Subsequently, in the tumor microenvironment, TGF-β1 promotes an immune escape process rendering NK cells inactive. In this regard, Lugini et al. [122] showed that resting and activated NK cells isolated from the blood of healthy donors released ExVs containing typical protein markers of NK cells, such as CD56^+^ and perforin. The ExVs exerted cytotoxic activity against different human tumor target cells and activated immune cells. On the other hand, fluorescence microscopy and flow cytometry indicated the different uptake efficiency of ExVs derived from different tumor cells by NK cells [123].

The MHC class I-related chain [MIC] A and MICB are ligands of NKG2D [124] and can be shed from tumor cells. The presence of these soluble molecules in blood is associated with a compromised immune response and progression of disease. Ashiru et al. [125] showed that treatment of NK cells with ExVs containing MICA not only downregulated NKG2D on the NK cell surface, but also provoked a marked reduction in NK cytotoxicity that was independent of NKG2D ligand expression by the target cell. Likewise, Lundholm et al. [126] found that ExVs secreted by prostate cancer cells expressed ligands for NKG2D on their surface and selectively downregulated NKG2D on NK and CD8^+^T cells, in a dose-dependent manner, leading to impaired cytotoxic function in vitro. Consistent with this observation, castration-resistant prostate cancer patients had significantly lower expression levels of NKG2D on circulating NK and CD8^+^T cells than healthy individuals. In these patients, tumor-derived ExVs seem also to be involved in the NKG2D downregulation since incubation of healthy lymphocytes with ExVs isolated from serum or plasma of the patients triggered downregulation of NKG2D expression in effector lymphocytes. In this context, Xiao et al. [127] investigated the regulatory effect of the histone deacetylase (HDAC) inhibitor drug MS-275 on hepatoma cell-derived ExVs, MICA and MICB. MS-275 modified hepatoma cell-derived ExVs carrying increased levels of HSP70 and MICB could significantly increase the cytotoxicity of NK cells and proliferation of PBMCs. 

In neuroblastoma, NK activation markers, such as NKG2D and DNAX-activating molecule (DNAM-1), correlate with low expression of the tumor-suppressive miR-186. Expression of the oncogenes MYC neuroblastoma (MYCN), aurora kinase A (AURKA), TGF-β receptors 1 and 2 is directly inhibited by miR-186. Neviani et al. [128] showed that NK cell-derived ExVs containing this miRNA exhibited cytotoxicity against MYCN-amplified neuroblastoma cell lines and prevented the TGF-β1-dependent inhibition of NK cells [128,129]. Moreover, Shoae-Hassani et al. [130] detected CD56, NKG2D and killer cell immunoglobulin like receptor [KIR] 2DL2 in ExVs derived from NK cells and exposed their cytotoxic effect on neuroblastoma cells. Conversely, they provided evidence that neuroblastoma-derived ExVs acted as tumor promoters by providing a tumor supporting niche. Zhu et al. [131] developed NK cell-derived ExV mimetics by extrusion of NK cells through filters with progressively smaller pore sizes and examined their anti-tumor effect. The anti-tumor activity of these mimetics was confirmed in a xenograft glioblastoma mouse model based on the significantly reduced size and weight of the tumor. Moreover, the cytotoxicity of mimetics against glioblastoma cells was associated with decreased levels of the cell survival markers p-ERK and p-AKT and with increased levels of the apoptosis protein markers cleaved-caspase 3, cytochrome-c and cleaved-PARP. This could suggest that these mimetics exerted stronger killing effects on cancer cells than the traditional NK-derived ExVs.

Treatment of multiple melanoma cells with sublethal doses of genotoxic drugs leads to senescence and increased NK cell recognition. Borrelli et al. [132] demonstrated that doxorubicin- and melphalan-treated senescent melanoma cells displayed elevated expression of the IL-15 receptor alpha subunit a complex on the cell membrane and IL-15 resulting in NK cell activation, proliferation and maturation. The drug-mediated stimulation was accompanied by increased release of ExVs that also expressed the IL15 receptor alpha subunit a complex. Their interaction with NK cells in the presence of exogenous IL15 resulted in increased NK proliferation. 

The dysfunction of NK cells mediated in an ExV-dependent manner was discovered by Xia et al. [133] in clear renal cell carcinoma (CRCC). ExVs from primary CRCC cells were preferentially enriched with TGF-β1 and mediated activation of the TGF-β/SMAD signaling pathway in NK cells resulting in functional NK deficiency. Moreover, the mediator role of tumor-derived ExVs in cholangiocarcinoma immune escape was documented by Chen et al. [134]. Cholangiocarcinoma cell-derived ExVs inhibited the antitumor activity of cytokine-induced NK cells by down-regulating the population of CD3^+^, CD8^+^, CD56^+^ and CD3^+^CD56^+^cells and the secretion of TNF-α and perforin. 

The relationship between hypoxic lung cancer-shed ExVs and NK-mediated cytotoxicity was investigated by Berchem et al. They found that ExVs derived from hypoxic cells qualitatively differed from those derived from normoxic tumor cells and inhibited more potently NK cell function. Following their uptake by NK cells, hypoxic lung cancer-derived ExVs released TGF-β1 in NK cells, decreasing the expression of NKG2D and consequently, inhibiting NK cell function. MiRNA profiling exposed the presence of high levels of miR-210 and miR-23a in these hypoxic ExVs. Since miR-23a directly targeted CD107a in NK cells, it might act as an additional immunosuppressive factor [135].

The stress-induced heat shock proteins (HSPs) are known to be involved in immunity and induce NK cell responses [136]. The immune regulatory effect of HSP-carrying ExVs secreted by hepatocellular carcinoma cells on NK cells was studied under stress conditions by Lv et al. [137]. Notably, cell-resistant anticancer drugs enhanced ExV release by HepG2 cells and generated more HSP-carrying ExVs which increased the activation of the cytotoxic response. The HSP-carrying ExVs stimulated NK cell cytotoxicity and granzyme B production, up-regulated the expression of inhibitory receptor CD94 and down-regulated the expression of activating receptors CD69, NKG2D, and NKp44.

Of note, Katsiougiannis et al. [138] isolated ExVs from salivary glands of mice and provided evidence that they exhibited a suppressive effect on pancreatic ductal adenocarcinoma [PDAC] resulting in a reduced tumor-killing capacity by NK cells.

## 6. Exosomes and Myeloid Cells 

### 6.1. Monocytes/Dendritic cells

Dendritic cells (DCs) are professional APCs that recognize, process and present antigens to T cells via MHC molecules [139]. Whereas immature DCs down-regulate T-cell responses to induce and maintain immunologic tolerance, mature DCs promote immunity. Depending on the maturation, DCs release ExVs with different cargos of miRNAs [140]. In addition, for the induction of antitumor immunity, DCs targeted by tumor-derived ExVs have been reported to be more effective for ExV-based vaccines [141]. In this regard, Romagnoli et al. [142] showed that sensitized T cells cultured with BC cells that were treated with DC-derived ExVs produced a significantly higher percentage of IFN-γ-secreting T cells for a potential immune response than did cells responding to untreated cells. Interestingly, Leone et al. [143] identified the lysosome-associated membrane protein LAMP-1 (CD107a) and LAMP-2 (CD107b) on the surface of DCs and showed that only LAMP-2 was internalized. As an endocytic receptor, LAMP-2 drove its cargo into an unusual Ag processing pathway, which reduced surface expression of Ag-derived peptides, but selectively enriching Ag within immunogenic ExVs. 

In 2006, Valenti et al. [144] provided the first evidence that tumor-released ExVs alter myeloid cell function by impairing monocyte differentiation into DCs and promoting the generation of a myeloid immunosuppressive cell subset. A subset of TGF-β-secreting CD14^+^HLA-DR monocytes that mediated suppressive activity on T lymphocytes was found to be significantly expanded in peripheral blood of melanoma patients compared with healthy donors. Furthermore, Wang et al. [145] described that MDSCs from multiple myeloma were able to take up ExVs derived from bone marrow stromal cells that induced their expansion in vitro. These ExVs activated the survival of the MDSCs through stimulating STAT3 and STAT1 pathways and increased the levels of anti-apoptotic proteins Bcl-xL and Mcl-1 [79]. The ExVs also elicited an increasing release of nitric oxide from multiple myeloma MDSCs so suppressing the activity of T cells [145]. On the other hand, the ability of ExVs derived from weakly metastatic melanoma cells and patients with non-metastatic primary melanomas to inhibit metastasis in the lung was shown by Plebanek et al. [146]. Due to the spreading of the lymphocyte antigen Ly6C^low^, the ExVs stimulated an innate immune response from patrolling monocytes in the bone marrow, which in turn caused cancer cell clearance at the pre-metastatic niche, recruiting of NK cells and TRAIL-dependent killing of melanoma cells by macrophages. This mechanism required the activation of the transcription factor Nr4a1 and was dependent on pigment epithelium-derived factor [PEDF] on the outer surface of ExVs. Likewise, Schuldner et al. [147] also investigated the inhibitory effect of melanoma-derived ExVs on lung metastasis mediated by the mobilization of Ly6C^low^. In this case, the formation of anti-tumor-ExVs was dependent on acetylation of p53 to elicit an immune response. In contrast, Wang et al. [148] showed that multiple melanoma-derived ExVs modulated the bone marrow microenvironment enhancing both angiogenesis and immunosuppression. Several pathways, such as STAT3, c-Jun N-terminal kinase and p53 were modulated in endothelial and bone marrow stromal cells by these ExVs. In naive mice, the ExVs promoted the growth of MDSCs by activating the STAT3 pathway and changing their phenotypes similar to those seen in multiple melanoma-bearing mice. 

In the blood circulation of glioblastoma patients CD14^+^CD163^+^M2-like monocytes, an indicator of Th2 bias, were elevated and associated with high serum concentrations of colony-stimulating factor (CSF) 2, CSF-3, IL-2, IL-4 and IL-13, as documented by Harshyne et al. [149]. A significant number of tumor ExV-reactive IgG2 and IgG4 antibody isotypes, connected with Th2 immunity, was also detected in the serum samples of these patients. Consistently, enriched serum ExVs were able to induce high expression levels of CD163 when added to normal monocytes. Conversely, Liu et al. [150] showed that orthotopic glioblastoma-bearing rats vaccinated with tumor-derived ExVs and α-galactosylceramide (α-GalCer)-pulsed DCs induced strong activation and proliferation of CTL, abrogating the immune tolerance. In addition, Bu et al. [151,152] generated ExVs from DCs loaded with chaperone-rich cell lysates (CRCLs) derived from glioma cells. These ExVs negatively regulated Casitas B cell lineage lymphoma (Cbl)-b and c-Cbl signaling, leading to the activation of phosphatidyl inositol 3-kinase (PI3K)/Akt and ERK signaling in T cells and thus, effectively induced anti-tumor T cell immune responses. The ability of hypoxia-stimulated glioma-derived ExVs to induce MDSCs was analyzed by Guo et al. [153] in vivo and in vitro. Hypoxia promoted the secretion of ExVs and mouse MDSCs took up the glioma-derived ExVs containing miR-10a and miR-21. The ExVs mediated expansion and activation of MDSCs by targeting RAR-related orphan receptor alpha (RORA) and PTEN. In line with this, mice injected with miR-10a- or miR-21-knockout glioma cells generated fewer MDSCs than those inoculated with wt glioma cells. Guo et al. [154] also showed that hypoxia-induced glioma cells stimulated the differentiation of MDSCs by transferring ExV miR-29a and miR-92a to MDSCs. MiR-29a and miR-92a activated the proliferation and function of MDSCs by targeting high-mobility group box transcription factor 1 (Hbp1) and protein kinase cAMP-dependent type I regulatory subunit alpha (Prkar1a), respectively. 

In the plasma of pancreas patients and xenograft cell lines, Javeed et al. [155] detected that ExVs decreased HLA-DR expression on CD14^+^monocytes and caused immune suppression in monocytes through altering STAT3 signaling and inducing arginase expression and reactive oxygen species. To increase the immune activity of ExVs for pancreas cancer, Que et al. [156] separated 128 ExV proteins, including several immune-activating proteins, and depleted ExV miRNAs. The anti-tumor effects of DC/cytokine-induced killer cells against pancreatic cancer was evaluated by proliferation and killing rates and TNF-α and perforin secretion. The researchers demonstrated that miRNA-depleted ExV proteins may be promising agonists to specifically activate DC/cytokine-induced killer cells against pancreas cancer. To determine the expression profiles of DCs treated with pancreas cancer-derived ExVs and normal DCs from healthy donors, Chen et al. [157] used integrated lncRNA and mRNA microarrays. LncRNAs, such as ENST00000560647 and mRNAs, such as legumain (lgmn) were differentially expressed in DCs treated with pancreatic cancer-derived ExVs. These ExV non-coding RNAs might play a critical role in immune escape because of their oncogenic function in DCs. Moreover, Zhou et al. [158] showed that pancreas cancer-derived ExVs downregulated TLR4 and downstream cytokines, such as TNF-α and IL-12, in DCs via miR-203.

ExVs that express HSP70 in their membrane are able to interact with and activate TLR2 on MDSCs. The antitumor effects of ExVs derived from hyperthermic CO_2_-treated DCs on gastric cancer cells was examined by Wang et al. [159]. In nude mice they decreased the expression of HSP70 and inhibited tumor growth. Likewise, Behzadi et al. showed that HSP70-enriched ExVs were an effective immunoadjuvant in cancer immunotherapy and caused fibrosarcoma tumor regression in a mouse model [160]. Finally, the significance of the peptide aptamer A8 in MDSC inhibition was reported by Gobbo et al. [161]. They perceived that this 8-aptamer bound to the extracellular domain of HSP70 and, therefore, used it to capture HSP70-expressing ExVs from breast, lung and ovarian cancer patient samples. Cisplatin and 5-fluorouracil increased the amount of HSP70 in ExVs, favoring the activation of MDSCs and impeding antitumor immune responses. In contrast, the MDSC activation was not observed if these chemotherapeutic agents were combined with A8, resulting in a strongly potentiated antitumor effect of the drugs.

Using a two-photon microscopy, Wang et al. [162] provided the first visual evidence on ovalbumin-specific DC-released ExVs targeting to cognate CD8^+^T cells via the ExV pMHC-I complex. They prepared HER2/neu-specific ExVs using adenoviral vector-transfected DCs and assessed their stimulatory effects on HER2/neu-specific CTL responses and antitumor immunity. These ExVs stimulated responses of CD8^+^T cells capable of not only inducing killing activity in HER2^+^melanoma and trastuzumab-resistant BC cells in vitro, but also eradicating 6-day palpable HER2^+^BC [3–4 mm in diameter] in athymic nude mice. Furthermore, Kitay et al. [163] showed that treatment of BC cells with the topoisomerase I inhibitor topotecan (TPT) induced DC activation via signaling of the cyclic GMP-AMP synthase (cGAS)-stimulator of interferon gene (STING), an important molecule in cytosolic DNA-mediated innate immune responses. Hence, the pathway drove antitumor immunity by responding to tumor cell-derived DNA. To complete these data, Diamond et al. [164] showed that tumor-derived ExVs secreted by irradiated mouse BC cells transferred dsDNA to DCs and upregulated the STING-dependent activation of IFN-I in DCs. In a prophylactic vaccination experiment, this event elicited CD8^+^T-cell responses and protected mice from tumor development significantly better than tumor-derived ExVs from untreated cancer cells. To increase their immune stimulatory ability and induce potent DCs, Taghikhani et al. [165] transfected tumor-derived ExVs isolated from the mouse BC cell line 4T1 with miR-155, miR-142 and let-7i by electroporation. They found that exosomal let-7i efficiently induced DC maturation, as identified by the maturation markers MHCII, CD80 and CD40.

To elicit T cell-mediated immune responses against HCC, Li et al. [166] loaded blood-derived DCs with a recombinant adeno-associated viral vector-carrying alpha-fetoprotein (AFP) gene and generated high-purity ExVs. They indicated that these ExVs effectively stimulated naive T cell proliferation and induced T cell activation to become antigen-specific CTL. Notably, DC precursors sensitized with these ExVs seemed to be more effective in triggering MHC I-restricted CTL response against HCC than mature DCs. As reported by Rao et al. [167], a significant inhibition of tumor growth could also be reached in ectopic and orthotopic HCC mice treated with DCs pulsed with tumor-derived ExVs. Orthotopic HCC mice treated with these DCs harbored an elevated number of T lymphocytes, increased levels of IFNγ and decreased levels of IL-10 and TGF-β at tumor sites. Irrespective of human leukocyte antigen types, a HCC-specific cytolysis elicited by DCs pulsed with HepG2 cell-derived ExVs was also observed across different HCC cell lines. In three different mouse models with ectopic, orthotopic or carcinogen-induced HCC tumors that displayed antigenic and pathological heterogeneity, Lu et al. [168] showed that ExVs derived from α-fetoprotein (AFP)-expressing DCs elicited strong antigen-specific immune responses resulting in a significant tumor growth retardation and prolonged survival. Significantly more IFN-γ-expressing CD8^+^T lymphocytes, elevated levels of IFN-γ and IL-2, fewer CD25^+^ forkhead box P3 (Foxp3) ^+^Tregs and decreased levels of IL-10 and TGF-β were detected at tumor sites.

In a phase II clinical trial on advanced NSCLC patients, Besse et al. [169] confirmed the capacity of a second generation of ExVs derived from IFN-γ-treated DC to boost the NK cell arm of antitumor immunity. The ExV levels of MHC class II correlated with those of the NKp30 ligand BCL2-associated athanogene 6 (BAG6) on the ExVs and with NKp30-dependent NK functions, the latter being associated with longer progression-free survival. In their study on lung cancer, Li et al. [170] demonstrated that NSCLC cells transfected with a vector expressing Rab27a [109] produced high levels of ExVs. These ExVs were able to upregulate MHC II molecules as well as the co-stimulatory molecules CD80 and CD86 on DCs, promoting the maturation process of DCs. In vitro, DCs loaded with these ExVs stimulated CD4^+^T cell proliferation, while in vivo immunization with them inhibited tumor growth in a mouse model whose splenocytes expressed high levels of type I cytokines. 

Conversely, Ning et al. investigated ExVs from Lewis lung carcinoma (LLC) as well as from BC cells and found that both ExV types blocked the differentiation of myeloid precursor cells into CD11c^+^DCs and induced cell apoptosis. In particular, the treatment with tumor-associated ExVs inhibited maturation and migration of DCs, promoted immune suppression of DCs, decreased CD4^+^IFN-γ^+^Th1 cell differentiation and increased the presence of Tregs. However, the immunosuppressive effect of DCs treated with tumor-derived ExVs could be partially restored by PD-L1 blockage [171].

High levels of ExVs containing miR-21 and miR-27 were measured in the plasma of patients with oral squamous cell carcinoma (OSCC) cancer by Momen-Heravi et al. Uptake of these ExVs by monocytes activated the NF-κB pathway and established a pro-inflammatory and pro-tumorigenic milieu associated with increased levels of IL-6, C-C motif chemokine ligand 2 (CCL2), prostaglandin E2 and matrix metallopeptidase MMP 9. Especially, these findings indicate the participation of ExV miR-21 in modulating the immune response in monocytes [172]. Likewise, lung cancer-derived ExVs induced DCs to produce IL-6 which promoted tumor invasion by increasing STAT3-dependent MMP 9 transcription activity in tumor cells [173]. Moreover, BC-derived ExV-mediated induction of IL-6 played a role in blocking BM DC differentiation [174].

Finally, Haderk et al. [175] showed that the transfer of plasma ExVs derived from chronic lymphocytic leukemia (CLL) contributed to cancer-related inflammation and concurrent immune escape via PD-L1 expression. In vitro and in vivo, they transported high levels of noncoding Y RNA hY4 to monocytes resulting in the release of cytokines, such as CCL2, CCL4, IL-6. These responses were abrogated in TLR7-deficient monocytes, suggesting ExV hY4 as a driver molecule of TLR7 signaling. Pharmacologic inhibition of endosomal TLR signaling resulted in a substantially reduced activation of monocytes in vitro and attenuated CLL development in vivo.

In cytotoxicity assays and animal studies, Yao et al. [176] revealed that DCs pulsed with leukemia cell-derived ExVs induced an antileukemic CTL immune response that was more effective than that of leukemia cell-derived ExVs and non-pulsed DCs, suggesting that leukemia cell-derived ExVs carry antigens and immunological molecules associated with leukemia cells. As reported by Huang et al., ExVs from TGF-β1 silenced murine leukemia L1210 cells were able to enhance the efficacy of DC-based vaccines. These ExVs significantly decreased TGF-β1 expression of DC and effectively promoted their maturation and immune function. In addition, DC pulsed with these ExVs more effectively favored CD4^+^T cell proliferation and Th1 cytokine secretion, induced anti-leukemic CTL response and inhibited tumor growth [177].

In their study on cervical cancer, Ren et al. [178] showed that CD1a^+^DCs loaded with HeLa-derived ExVs promoted T cell proliferation and induced cytotoxic responses to inhibit the growth of cervical cancer cells in vitro. Chen et al. [179] showed that DC-derived ExVs loaded with the immuno-dominant E7 epitope (E7_49–57_) efficiently induced the cytotoxic activity of CD8^+^T cells against TC-1 tumor cells and their proliferation and IFN-γ excretion. Incidentally, the TLR-3 ligand poly(I:C) dramatically increased the potent antitumor immunity against cervical cancer induced by ExVs derived from antigen-pulsed DCs. In cervical cancer-bearing mice, DC-derived ExVs loaded with poly(I:C) and E7 markedly inhibited tumor growth and improved the survival rate. Damo et al. [180] also used poly(I:C), however, together with ovalbumin to produce DC-derived ExVs that strongly stimulated ovalbumin-specific CD8^+^ and CD4^+^T cells. When a melanoma cell lysate was used to load DCs with tumor antigens together with poly(I:C), ExVs were generated, capable of inducing activation of melanoma-specific CD8^+^T cells and recruiting cytotoxic CD8^+^T cells, NK and NK-T cells to the tumor site, resulting in significantly reduced tumor growth and enhanced survival.

### 6.2. Macrophages

M2 macrophages have been reported to promote tumor progression, angiogenesis and metastasis through regulating T-cell function. Whereas TAMs are largely immunosuppressive, their ExVs seem to have the potential to stimulate, rather than suppress, anti-tumor immunity [181].

Using quantitative proteomics, Park et al. [182] identified immunomodulatory proteins and chemokines, including CSF-1, CCL2, FTH, FTL and TGF-β, in ExVs produced by diverse hypoxic tumor cells. The hypoxia-induced vesicles were able to influence macrophage recruitment and promote M2-like polarization both in vitro and in vivo. In addition, hypoxic, but not normoxic, melamona-derived ExVs enhanced oxidative phosphorylation in bone marrow-derived macrophages through transferring let-7a, resulting in the inhibition of the insulin-Akt-mTOR signaling pathway.

In HCC, Li et al. [183] suggested a critical function of ExV lncRNA TUC339 in the regulation of macrophage M1/M2 polarization. HCC-derived ExVs contained elevated levels of TUC339 and were taken up by ThP-1 cells. Following suppression of TUC339, the researchers observed increased pro-inflammatory cytokine production and phagocytosis in THP-1 cells with the opposite effect upon over-expression of this lncRNA. Higher levels of TUC339 in M(IL-4) macrophages than in M(IFN-γ^+^LPS) macrophages and a down-regulated TUC339 expression during M(IL-4)-to-M(IFN-γ^+^LPS) repolarization were also detected. Applying a microarray, the researchers identified cytokine-cytokine receptor interaction, CXC chemokine receptor binding, Toll-like receptor signaling, FcγR-mediated phagocytosis, regulation of the actin cytoskeleton, and cell proliferation to be associated with TUC339 function in macrophages as well as with HCC growth and spread. Cheng et al. [184] treated HCC cells with melatonin and detected that the secreted ExVs downregulated the expression of PD-L1 on macrophages and attenuated the expression of IL-6, IL-10, IL-1β and TNF-α so reducing the activation of STAT3, whereas control ExVs derived from untreated HCC cells upregulated the secretion of these inflammatory cytokines in macrophages. 

In BC, Chow et al. [185] demonstrated that only BC-, but not non-cancer cell-derived ExVs, stimulated NF-κB activation in macrophages resulting in secretion of IL-6, TNFα, GCSF and CCL2. Palmitoylated proteins present on the surface of BC-derived ExVs contributed to the NF-κB activation. In mice bearing xenografted BC, BC-derived ExVs were internalized by macrophages in axillary lymph nodes thereby triggering IL-6 expression. Genetic ablation of TLR2 or myeloid differentiation factor 88 (MyD88) completely abolished the activating effect of tumor-derived TxVs. In contrast, inhibition of TLR4 or endosomal TLRs (TLR3/7/8/9) failed to abrogate the NF-κB activation by ExVs. Conversely, Jang et al. [186] showed that the green tea constituent epigallocatechin gallate (EGCG) suppressed tumor growth in a murine BC model that was associated with decreased TAM and M2 macrophage infiltration. In BC cells from EGCG-treated mice, expression of CSF-1 and CCL-2 were low. As verified by decreased IL-6 and TGF-β and increased TNF-α levels, EGCG skewed cytokines of TAM from M2- into the M1-like phenotype. EGCG also up-regulated miR-16 which was transferred to TAM by ExVs and inhibited TAM infiltration and M2 polarization.

The reprogramming of macrophages by mutant TP53 cancer cells was reported by Cooks et al. [187]. Colon cancer cells harboring gain-of-function mutant p53 selectively secreted ExVs enriched with miR-1246. Uptake of these ExVs by neighboring macrophages triggered a miR-1246-dependent reprogramming which promoted anti-inflammatory immunosuppression with an increased TGF-β signaling activity. Firstly, Wang et al. [188] indicated that GC-derived ExVs induced monocytes to differentiate into PD1^+^ TAMs with M2 phenotype. Secondly, in advanced GC, they identified a pro-tumorigenic subset of macrophages that constitutively expressed PD1. These PD1^+^TAMs exhibited an M2-like surface profile with an increased expression of CD206, IL-10 and CCL1 and a decreased expression of MHC II, CD64 and IL-12 as well as the ability to phagocytose ovalbumin and to suppress CD8^+^T-cell function. In glioblastoma, Gabrusiewicz et al. [189] showed that ExVs containing members of the STAT3 pathway traversed the monocyte cytoplasm, caused a reorganization of the actin cytoskeleton and shifted monocytes towards the immunosuppressive M2 phenotype. This event was associated with PD-L1 expression. In head and neck cancer, Hsieh et al. [190] detected a high expression of miR-21 which was associated with higher levels of the EMT transcription factor SNAI1 and the M2 marker MRC1. SNAIl activated the transcription of miR-21 that was integrated into ExVs. CD14^+^monocytes that took up these ExVs suppressed the expression of M1 markers and increased M2 markers. In co-culture experiments, Kanlikilicer et al. [191] demonstrated that ExVs released from OC cells transported oncogenic miR-1246 to M2-type macrophages, but not to M0-type macrophages. In this process of ExV transfer, the integral membrane protein caveolin-1 gene, a direct target of miR-1246, was involved. 

In in vitro and in vivo assays, M1 macrophage-derived ExVs containing miR-16-5p triggered a T cell immune response that in turn inhibited tumor formation by decreasing the expression of PD-L1. In HCC an interaction of lncRNA PCED1B-AS1 with miR-194-5p was observed by Fan et al. [192]. Yao et al. [193] reported that ExV miR-27a-3p promoted immune evasion by up-regulating PD-L1 via MAGI2/PTEN/PI3K axis in breast cancer. In mechanistic studies, coculture of breast cancer cell derived ExVs with macrophages suggested that endoplasmic reticulum stress biomarkers including GRP78, PERK, ATF6, IRE1α and PD-L1 were overexpressed in breast cancer tissues relative to para-cancerous tissues. Endoplasmic reticulum stress promoted ExV secretion. In macrophages, elevated levels of miR-27a-3p and PD-L1 were observed in response to ExV-overexpressing miR-27a-3p in vivo and in vitro. MiR-27a-3p targeted and negatively regulated MAGI2, while MAGI2 down-regulated PD-L1 by up-regulating PTEN to inactivate the PI3K/AKT signaling pathway. Less apoptosis occurred in CD4^+^, CD8^+^ T cells and IL-2, and T cells as a response to the co-culture of macrophages and CD3^+^ T cells. 

Applying experimental tumor implantation, Guo et al. [194] found that mitogen-activated protein kinase 2 [MEKK2]-deficient mice were more resistant to viral infection than wild-type mice. With regard to this observation, they showed that Lewis lung cancer cells were able to repress innate antiviral immunity of macrophages by transferring EGFR^+^ExVs. Active EGFR stimulated MEKK2 that, in turn, phosphorylated IRF3 so leading to repression of IRF3 and IFN-β1 and weakening of the pathogen-defense ability of macrophages.

## 7. Exosomes and Cancer Therapy

Although ExVs have a relative longevity within the blood circulation and an ability to cross the blood–brain barrier [195], modified ExVs appear to only spend a short time in blood circulation [196]. ExVs encapsulating anti-inflammatory drugs have been used by Zhuang et al. [195] in the treatment of brain inflammation and introduced as a means of treatment via the nose. Untreated ExVs purified from seven day autologous derived-DC monocyte cultures have also been used via four vaccinations to each of 15 tumor patients. A partial response was exhibited by one patient in the tumor bed associated with progressive loss of HLA-A2 and HLA-BC molecules on tumor cells while one minor, two stable and one mixed response occurred in skin and lymph node sites [197].

Clearly, modified ExVs offer the potential to create new biological tools for cancer therapy. Hence various technologies have been proposed to load ExVs with exogenous cargoes [198]. One such approach concerns the transfection of purified ExVs by the use of electroporation. In this case, ExVs are suspended in a conductive solution. An electrical pulse at an optimized voltage, lasting only a few microseconds to a millisecond, is discharged through the ExVs. This leads to the formation of temporary pores in the ExV membrane. The rise in the electric potential across the ExV membrane allows charged molecules e.g., DNA and RNA to cross the membrane [199]. Hence, it is possible to load ExVs with miRNAs e.g., miR-21 [200,201]. 

Other distinct approaches to load ExV carriers with a therapeutic cargo include the loading of naïve ExVs that have been isolated from parental cells ex-vitro and the loading of parental cells with a drug that is subsequently released via the ExVs. Batrakova and Kim [202] have suggested that each approach, together with electroporation, may have advantages and limitations dependent upon the disease site, the type of therapeutic cargo and the most suitable conditions relevant to a specific type of ExV-encapsulated cargo. Alternatively, either the loading of donor cells with drugs or the transfection of donor cells with DNA encoding for a drug could be employed with the modified EVs released from such cells being extracted from the culture medium [202].

A brief review of the use of breast cancer cell, dendritic, mesenchymal cell and macrophage derived ExVs in both direct effects on tumor cells and through their use as delivery systems were considered by Schwarzenbach and Gahan [21].

In addition, ExVs have also been used for the insertion of abiotic catalysts into cancer cells, i.e., the use of ex-directed catalyst prodrug therapy. Thus, transition metals, such as palladium (Pd), rubidium (Ru) and gold (Au) have been inserted into living cells, tissues and animals in order to find ways of catalyzing first-in-life reactions. A selected number of agents such as organometallic complexes, artificial metalloenzymes and metal-loaded nanocarriers have been used to demonstrate that they can maintain their catalytic activity and their functional compatibility within living cells. Various examples have been reported to determine the feasibility of this concept [203,204]. Thus, Pérez-Lopez et al. [203] demonstrated that solid-supported gold nanoparticles were able to uncage pro-both FudR (floxuridine) and POB-SAHA BOP (Bioorthogonal Oxygen-Independent Prodrug Strategy suberanilohydroxamic acid) while Sancho-Albero et al. [204] have developed a bio-artificial device involving cancer cell ExVs loaded with Pd catalysts. The method employed results in a controlled assembly of Pd nanosheets inside the ExVs as well as mediating Pd-triggered dealkylation reactions inside cells. Importantly, there is a preference for similar cell types to those from which the exosomes originated. This manner of delivering abiotic catalysts into designated cancer cells offers a new targeted therapy permitting the discrimination between tumor cell types. Thus, during the use of panobinostat, intracellular catalytic properties of Pd were demonstrated in lung cancer A549 cells using ExVs from these cells. However, they had no such effects on cells from the U87 glioma cell line and vice versa [204]. Pd has also been exploited to uncage the active metabolites of irinotecan [205] as well as the cytotoxic paclitaxel [206].

Chronopoulos and Kallur [207] have considered emerging roles for the use of ExVs from bacterial sources [BEVs], suggesting, in particular, their possible roles in liquid biopsies and bioengineering strategies for cancer therapy. Thus, they indicate that microbial digestive disturbances could trigger the release of BEVs acting to promote tumors via suppressing monocytic differentiation in a TLR-dependent fashion leading to T-cell anergy [208]. They also suggest that BEV-associated with microbe associated molecular patterns could affect distant organs through the initiation of pro-inflammatory signals leading to the preparation of future metastatic niches.

If natural sources of ExVs are to be employed, as out-lined above, then their safe clinical use must be verified. Moreover, the avoidance of any interactions between therapeutic ExVs and unintended cells needs also to be examined [23,204]. The study of Wu et al. [23] indicated that standard guidelines for the manufacture, purification, storage, usage, duration and dosage of ExV-based drugs need to be established. 

Ideally, the construction of artificial ExVs carrying the required molecule[s] could be an advantage and eliminate possible unwanted outcomes due to nucleic acids, proteins and lipids already present as ExV cargo. Such ExVs need to be empty and to permit [a] the addition of the selected immune derived molecule or anti-tumor compound and [b] to have surface markers to direct them to engage with a particular cell type. A further advantage in the case of an anti-tumor compound is the relatively low dosage needed to act efficiently via ExVs rather than a higher dosage via a whole body dose. As well as reducing the treatment load of the patient, there is a cost advantage when using expensive molecules.

Thus, De La Pena et al. [209] prepared FDA (Food and Drug Administration) approved liposomes coated with an optimized number of MHC I peptide complexes and a selected specific range of ligands for adhesion, early activation, late activation and survival TCR. They were able to follow the movement of such artificial ExVs via either fluorescence or magnetic resonance imaging and to demonstrate that artificial ExVs both activated and expanded functional antigen specific T cells. Equally, the use of those ExVs secreted by tumor cells can be exploited to ensure specificity of binding to the similar tumor cell type [204].

## 8. Conclusions

There is clear preliminary evidence that ExVs, both natural and artificial, offer the possibility of the delivery of therapeutic immunology components for the control and destruction of tumor cells. The variety of approaches employing ExVs described in the literature offers a variety of possible treatment regimes.

The release of ExVs by all cells within the tumor microenvironment e.g., endothelial cells, tumor-associated fibroblasts and immune system cells leads to the movement of their cargo from e.g., tumor cells to healthy cells and vice versa from healthy to tumor cells. The information carried via the proteins, nucleic acids and lipids present therein can lead to e.g., tumor cell invasion and immune escape. Moreover, ExVs can also suppress tumor progression as a result of antigen presentation to immune cells since tumor-derived ExVs carry surface antigens similar to those of the donor cells. Conversely, dendritic cell-derived ExVs contain MHC I and II peptide complexes so having the capability of priming other immune system cell types leading to the activation of an antitumor immune response [210].

Although ExVs have a relative longevity within the blood circulation and an ability to cross the blood–brain barrier [195], modified ExVs introduced as a means of treatment appear to only spend a short time in circulation [196]. However, modified organelles offer the potential to create new biological-tools for cancer therapy. Hence various technologies have been proposed to load ExVs with exogenous cargoes [198], such as the use of electroporation with purified ExVs. Alternative approaches involve either the loading of donor cells with drugs or the transfection of donor cells with DNA encoding for a drug. Modified ExVs released from such cells are then extracted from the culture medium [202]. ExVs have also been used for the insertion of abiotic catalysts into cancer cells, i.e., the use of ExV-directed catalyst prodrug therapy [204]. If natural sources of ExVs are employed then their safe clinical use must be verified. Moreover, the avoidance of any interactions between therapeutic ExVs and unintended cells needs also to be examined [23]. This study suggested that standard guidelines for the manufacture, purification, storage, usage, duration and dosage of ExV-based drugs need to be established. 

Ideally, the construction of artificial ExVs carrying the required molecule[s] could be an advantage and eliminate possible unwanted outcomes due to nucleic acids, proteins and lipids already present as ExV cargo. Such ExVs need to be empty and to permit [a] the addition of the selected immune derived molecule or anti-tumor compound and [b] to have surface markers to direct them to engage with a particular cell type. A further advantage in the case of an anti-tumor compound is the relatively low dosage needed to act efficiently via ExVs rather than higher dosage via a whole body dose. As well as reducing the treatment load of the patient, there is a cost advantage when using of expensive molecules.

Thus, De La Pena et al. [209] prepared FDA approved liposomes coated with an optimized number of MHC I peptide complexes and a selected specific range of ligands for adhesion, early activation, late activation and survival TCR. They were able to follow the movement of such artificial ExVs via either fluorescence or magnetic resonance imaging, and to demonstrate that artificial ExVs both activated and expanded functional antigen specific T cells. 

There are many similarities between ExVs and liposomes having resulted in liposomes being the favored type of nanoparticle to replace actual ExVs as drug carriers [196]. In addition, on being pegylated, liposomes can circulate for longer periods than applied ExVs. An alternative approach proposed by Zhang et al. [211] involved the development of artificial chimeric ExVs (ACEs). These are assembled from cell membrane proteins isolated from a variety of cell types into synthetic phospholipid bilayers. The ACEs showed, in vivo, a greater tumor accumulation, lower interception and better antitumor therapeutic effect than standard liposomes.

Nanoparticles could offer a very effective alternative to either ExVs or liposomes for cancer immunotherapy due to their high specificity, efficacy, ability to diagnose, imaging and therapeutic effect [212]. A range of nanoparticle systems including polylactic-co-glycolic acid nanoparticles, liposomes, micelles, gold nanoparticles, iron oxide, dendrimers may be used independently in conjunction with artificial ExVs widely used for immunotherapy of cancer. 

## Figures and Tables

**Figure 1 ncrna-07-00004-f001:**
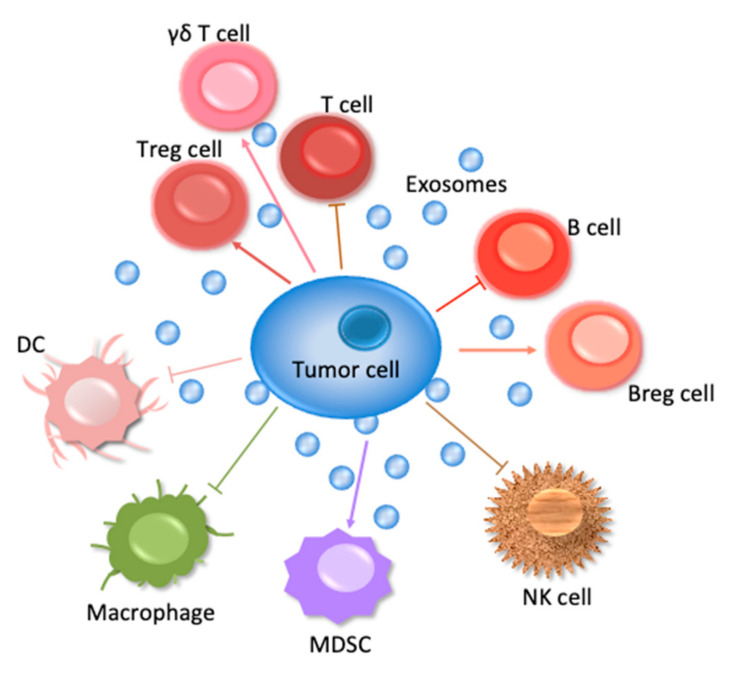
Interactions of immune cells with a tumor cell. Lymphoid cells comprise T, B, NK cells, and myeloid cells comprise DC, macrophages.

**Table 1 ncrna-07-00004-t001:** ExVs carrying miRNAs, lncRNAs or PD-L1 in immune regulation.

miRNA	ExV Release	Induction	Tumor	Reference
miR-155	bystander T-cell activated DCs	Antigen-specific T cells	AML	[52]
let-7i	cells	Expression of MHCII, CD80 and CD40, DC maturation,	breast	[53]
miR-16	EGCG-treated cells	Inhibition of TAM infiltration and M2 polarization.	breast	[54]
miR-92	CAFs	higher cellular levels of PD-L1 and miR-92; apoptosis, impaired proliferation of T cells	breast	[55]
miR-27a-3p	cells	immune evasion by up-regulating PD-L1 via MAGI2/PTEN/PI3K axis in macrophages	breast	[56]
miR-15a, miR-15b, miR-16, miR-193a-3p		PD-L1 targets	BE, EAC	[57]
miR-1246	mutant TP53 cells	Increase in TGF-β, reprogramming of macrophages	colon	[58]
miR-29a-3p, miR-21-5p	TAMs	Suppression of STAT3 in CD4^+^T cells	EOC	[59]
miR-101	cells	Targeting of EZH2, EED, DNMT3, reversion of promoter methylation, re-expression of LRRC4 in GBM cells	GBM	[60]
miR-16-5p	M1 macrophage	T cell immune response that in turn inhibits tumor formation by decreasing the expression of PD-L1	GC	[61]
miR-10a, miR-21	hypoxic cells	Targeting of RORA and PTEN Expansion and activation of MDSCs	glioma	[62]
miR-29a, miR-92a	hypoxic cells	Targeting of Hbp1 and Prkar1a, differentiation of MDSCs	glioma	[63]
miR-21	cells	Suppression of M1, increase in M2 markers in CD14^+^monocytes	HNC	[64]
miR-690	melanoma	Activation of caspase-3, -7 -9, down-regulation of BCL-2, MCL-1, BCL-xL in CD4^+^T cells	melanoma	[65]
miR-150-5p, miR-142-3p	Tregs	Increase in IL-10, decrease in IL-6 following LPS stimulation in DCs	mice	[66]
miR-186	NK cells	Cytotoxicity against neuroblastoma cell, inhibition of NK cells	neuro-blastoma	[67,68]
miR-24-3p	serum	repression of FGF11, Suppression of T-cells	NPC	[69]
miR-24-3p, miR-891a, miR-106a-5p, miR-20a-5p, miR-1908,	cells	Down-regulation of MARK1 signaling pathway in T cells	NPC	[70]
miR-210, miR-23a	hypoxic cells	Release of TGF-β1 in NK cells, decrease in expression of NKG2D, inhibition of NK cell function	NSCLC	[71]
miR-320d, miR-320c, miR-320b miR-125b-5p	cells	efficacy of immunotherapy	NSCLC	[72]
miR-1246	cells	transfer to M2-type macrophages, but not to M0-type macrophages	OC	[73]
miR-21, miR-27	plasma	Activation of NF-κB pathway increase in IL-6, CCL2, prostaglandin E2 and MMP 9 in monocytes	OSCC	[74]
miR-203	cells	Downregulation of TLR4, TNF-α and IL-12 in DCs	pancreas	[75]
**lncRNA**				
PCED1B-AS1	cells	Interaction of PCED1B-AS1 with miR-194-5p, increased PD-L1 and PD-L2 expression, inhibition of T cells and macrophages, cell proliferation, colony formation, inhibited apoptosis, tumor formation	HCC	[76]
TUC339	cells	Toll-like receptor signaling, regulation of the actin cytoskeleton, regulation of macrophage M1/M2 polarization	HCC	[77]
MALAT1, HOTAIR, HOTTIP, AGAP2-AS1, ATB, TCF7, FOXD2-AS1, HOXA11-AS, PCAF1, BCAR4	cells	Cancer development	NSCLC	[78]
ENST00000560647	cells	Differential expression in DCs, immune escape	pancreas	[79]
**PD-L1**				
	draining lymph node	Suppression of T cell activation		[51]
	HER2-positive cells	immunosuppressive	breast	[80]
	cells	Inhibition of maturation and migration of DCs, decrease in CD4^+^IFN-γ^+^Th1 cell differentiation, increase in Tregs.	breast, LLC,	[81]
	cells	Inflammation, immune escape	CLL	[82]
	cells	Shift of monocytes towards M2 phenotype.	glioblastoma,	[83]
	cells	Secretion of IL-6, IL-10, IL-1β and TNF-α, reduction of activation of STAT3 in macrophages	HCC	[84]
	cells	immune cell reprogramming by the adenosine pathway	HNSCC	[85]
	cells	Suppressing of CD8^+^T cells, positive correlation with IFN-γ	metastatic melanoma	[86]
	plasma	Reduction of the secretion of IFN-γ by T cells, apoptosis in CD8^+^T cells	NSCL	[87]

## Data Availability

No new data were created or analyzed in this study. Data sharing is not applicable to this article.

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
