# Peer review of "Exosomes in Immune Regulation"

_ncrna, 2021, doi:10.3390/ncrna7010004_

Round 1

Reviewer 1 Report

The authors discussed the role of exosomes in immunomodulation of lymphoid and myeloid cells, and their use as immune stimulatory agents to elicit specific cytotoxic responses against the tumor. Generally, this review is well written and summarized recent findings regarding exosomes and immune regulation. However, there are still some minor points need to be addressed before publication.

a) The biogenesis process of ExVs is suggested to be highlighted and separated from section 1 (line 18, introduction). Current ExVs isolation and characterization methodology can be added in introduction.

b) The inner logic between section 2 (line 71), 3 (line 82) and 4 (line 125) is unclear and confusing. The authors can merge sections 3 and 4 under section 2.

c) Add lymphoid cells (T, B, NK cells) and myeloid cells (DC, macrophages) in Figure 1.

d) Better to create a table to display the contents in section 6 (line 155) and section 7 (line 518).

e) For better understanding, "exosomes" and "ExVs" can be unified.

f) Some sentences need to be simplified. e.g. line 28-30.

Author Response

Manuscript ID: ncrna-1013556

Title Exosomes in immune regulation

Authors Heidi Schwarzenbbach *  Peter B. Gahan

Point-by-point details of the revisions in the manuscript and responses to the reviewers' comments. Our answers are in italic. The revisions are highlighted in red in the manuscript. The authors thank the reviewers for their helpful comments and time.

a) The biogenesis process of ExVs is suggested to be highlighted and separated from section 1 (line 18, introduction). Current ExVs isolation and characterization

methodology can be added in introduction.

The biogenesis process of ExVs was separated from section 1 (pages 2,3) and ExVs isolation and characterization methodology was added to the introduction (pages 1,2).

b) The inner logic between section 2 (line 71), 3 (line 82) and 4 (line 125) is unclear and confusing. The authors can merge sections 3 and 4 under section 2.

We now clarified section 2 (pages 2-4)

c) Add lymphoid cells (T, B, NK cells) and myeloid cells (DC, macrophages) in Figure 1.

We now added that lymphoid cells comprise T, B, NK cells, and myeloid cells comprise DC, macrophages (page  5).

d) Better to create a table to display the contents in section 6 (line 155) and section 7 (line 518).

A Table 1 was created to display the contents in section 6 and section 7 (pages 5-8).

e) For better understanding, "exosomes" and "ExVs" can be unified.

Now, exosomes have been abbreviated Ms throughout the manuscript. Changes are in red.

f) Some sentences need to be simplified. e.g. line 28-30.

The sentences were simplified (in red).

Reviewer 2 Report

In the article entitled “Exosomes in immune regulation” Heidi Schwarzenbach and  Peter B. Gahan in interesting form wrapped up extremely important aspect of the effect of extracellular vesicles on cells of the immune system. I believe that witnessing how mRNA-based vaccine encapsulated within lipid vesicle mobilize immune cells to produce antibodies, will significantly impact extracellular vesicle community and expand ncRNA-vesicle therapy tests.

So I encourage the authors to rewrite manuscript to address the RNA role directly. The concept that cellular ncRNA expression change composition of the cancer EV  have a range of examples at disposal.   Discussion of recent studies by authors colleague  from Center Hamburg-Eppendorf showing cancer cell dependent PDL1 presentation on EVs or lncRNA based deregulation of PDL1 would also be beneficial.

Author Response

Manuscript ID: ncrna-1013556

Title Exosomes in immune regulation

Authors Heidi Schwarzenbbach *  Peter B. Gahan

Point-by-point details of the revisions in the manuscript and responses to the reviewers' comments. Our answers are in italic. The revisions are highlighted in red in the manuscript. The authors thank the reviewers for their helpful comments and time.

So I encourage the authors to rewrite manuscript to address the RNA role directly. The concept that cellular ncRNA expression change composition of the cancer EV have a range of examples at disposal. Discussion of recent studies by authors colleague from Center Hamburg-Eppendorf showing cancer cell dependent PDL1 presentation on EVs or lncRNA based deregulation of

PDL1 would also be beneficial.

We thank the referee for the interesting suggestion to write a review that addresses the RNA role directly. Regretfully, we are unable to do this at this point in time. Moreover, we wished to review “Exosomes in immune regulation” on this occasion. We prefer to offer a corrected version following the first referee’s comments since the manuscript that has already been accepted for publication by the first referee and the Editor.

However, we have added some important aspects of ncRNAs and PDL1 to the manuscript (pages 9, 10, 20, 21) and Table 1 (pages 5-8). 

We trust that you will support this decision.

Round 2

Reviewer 1 Report

Agree to publish.

Reviewer 2 Report

I agree with authors revision.